Sexual coevolution of spermatophore envelopes and female genital traits in butterflies: Evidence of male coercion?

Sánchez Víctor 1
Cordero Carlos 2 cordero@ecologia.unam.mx
1 Posgrado en Ciencias Biológicas, Instituto de Ecología, Universidad Nacional Autónoma de México , México , Distrito Federal , Mexico
2 Departamento de Ecología Evolutiva, Instituto de Ecología, Universidad Nacional Autónoma de México , Distrito Federal , Mexico
Huber Dezene
Electronic publication date: 2014 Jan 30
Publication date: 2014
Volume: 2
Electronic Location ID: e247
Received 2013 Oct 27; Accepted 2013 Dec 29
Copyright: © 2014 Sánchez et al.
Copyright year: 2014
Copyright holder: Sánchez et al.
License: This is an open access article distributed under the terms of the Creative Commons Attribution License, which permits unrestricted use, distribution, and reproduction in any medium, provided the original author and source are credited.
License URL: https://creativecommons.org/licenses/by/3.0/

Keywords: Sexual coevolution, Sexual selection, Sexual Conflict, Female genitalia, Spermatophore, Signa, Lepidoptera, Heliconiinae, Sexual coevolution

Funding: This research was supported by PAPIIT/DGAPA IN208413 (UNAM), the Instituto de Ecología (UNAM), and the Posgrado en Ciencias Biológicas (UNAM). Víctor Sánchez received a doctoral scholarship from CONACYT. The funders had no role in study design, data collection and analysis, decision to publish, or preparation of the manuscript.

==============================
Signa are sclerotized structures located on the inner wall of the corpus bursa of female Lepidoptera whose main function is tearing open spermatophores. The sexually antagonistic coevolution (SAC) hypothesis proposes that the thickness of spermatophore envelopes has driven the evolution of the females signa; this idea is based in the fact that in many lepidopterans female sexual receptivity is at least partially controlled by the volume of ejaculate remaining in the corpus bursa. According to the SAC hypothesis, males evolved thick spermatophore envelopes to delay the post-mating recovery of female sexual receptivity thus reducing sperm competition; in response, females evolved signa for breaking spermatophore envelopes faster, gaining access to the resources contained in them and reducing their intermating intervals; the evolution of signa, in turn, favored the evolution of even thicker spermatophore envelopes, and so on. We tested two predictions of the SAC hypothesis with comparative data on the thickness of spermatophore envelopes of eleven species of Heliconiinae butterflies. The first prediction is that the spermatophore envelopes of polyandrous species with signa will be thicker than those of monandrous species without signa. In agreement with this prediction, we found that the spermatophore envelopes of a polyandrous Heliconius species with signa are thicker than those of two monandrous Heliconius species without signa. The second prediction is that in some species with signa males could enforce monandry in females by evolving “very thick” spermatophore envelopes, in these species we predict that their spermatophore envelopes will be thicker than those of their closer polyandrous relatives with signa. In agreement with this prediction, we found that in two out of three comparisons, spermatophore envelopes of monandrous species with signa have thicker spermatophore envelopes than their closer polyandrous relatives with signa. Thus, our results support the idea that selective pressures arising from sexually antagonistic interactions have been important in the evolution of spermatophore envelopes, female signa and female mating patterns.

Introduction

During sexual interactions males and females exert selection pressures on the opposite sex that can produce reciprocal adaptations in a process known as sexual coevolution (Parker, 1979; Eberhard, 1985, 1996; Holland & Rice, 1998). There is increasing evidence that sexual coevolution is responsible for the evolution of many structural and functional aspects of animal genitalia (Eberhard, 1985, 1996, 2010; Hosken & Stockley, 2004; Arnqvist & Rowe, 2005; Minder, Hosken & Ward, 2005; Brennan et al., 2007; Sánchez, Hernández-Baños & Cordero, 2011; Breed, Leigh & Speight, 2013; Burns, Hedin & Shultz, 2013; Yassin & Orgogozo, 2013). For example, in species in which females increase their fitness by mating with multiple mates, males could evolve genital structures for damaging female genitalia if this damage decreases female mating rates; these structures, in turn, could select for protective genital structures in females. In the Drosophila melanogaster species subgroup evidence indicates that females have coevolved genital structures that protect them from damage by male genital structures (Yassin & Orgogozo, 2013). In other species, also exhibiting adaptive polyandry, females could evolve genital traits that allow them to discriminate among males of different quality during copulation; these traits could select for elaborate male intromittent genitalia for internal stimulation of the females (Eberhard, 1985). Evidence suggests that the extremely complex vaginal morphology of waterfowl species coevolved with the long and complex male phallus as a cryptic choice mechanism (Brennan et al., 2007).

In the particular case of Lepidoptera, in a previous paper we presented comparative evidence supporting the hypothesis that the sclerotized structures called signa, present in the inner genitalia of females from many species, are a product of antagonistic coevolution with males (Sánchez, Hernández-Baños & Cordero, 2011). The signa are located on the inner wall of the female’s corpus bursa—the bag-like receptacle where males deposit a spermatophore during copulation—and are used for breaking the spermatophore envelope and gain access to its contents (Hinton, 1964; Galicia, Sánchez & Cordero, 2008; Lincango, Fernández & Baixeras, 2013). Our sexually antagonistic coevolution (SAC) hypothesis proposes that, since in many polyandrous Lepidoptera the length of time a female remains sexually unreceptive after mating is directly related to the amount of ejaculate remaining in her corpus bursa (Sugawara, 1979; Drummond, 1984; Wiklund, 2003; Wedell, 2005), sperm competition selects for males that transfer spermatophores with thick envelopes that take more time to be broken and thus delay female remating beyond her optimum time (Drummond, 1984; Cordero, 2005; Fig. 1). Thick spermatophore envelopes, in turn, select for signa that allow females faster breaking of the envelopes, thus reducing intermating intervals (Cordero, 2005; Fig. 1). Our previous comparative analysis supported the prediction from the SAC hypothesis that signa tend to be present mainly in polyandrous species, and suggested that polyandry and signa are plesiomorphic in the Lepidoptera (Sánchez, Hernández-Baños & Cordero, 2011). The SAC hypothesis also predicts that when monandry is selected for in females, the resulting disappearance of sperm competition favors the evolution of thinner spermatophore envelopes (because they are less expensive to produce) and, in consequence, the loss of signa in females. Our previous study also found support for this prediction, because in several groups (including the pupal mating Heliconius species) the evolution of monandry was accompanied by the loss of signa (Sánchez, Hernández-Baños & Cordero, 2011). However, in some cases monandry could be imposed by males on females (i.e., could be maladaptive for females) by evolving even thicker spermatophore envelopes in response to the evolution of signa (an analogous effect has been proposed for Heliconius antiaphrodisiacs; Estrada et al., 2011). In this case, the SAC hypothesis predicts the evolution of thicker spermatophore envelopes in monandrous species with signa than in polyandrous species.

Figure 1 Sexually antagonistic coevolution hypothesis of the evolution of spermatophore envelope thickness and signa in Lepidoptera.

Schematic representation of the Sexually Antagonistic Coevolution hypothesis for the coevolution of spermatophore envelopes and signa in Lepidoptera. Arrows represent selective pressures.

Predictions of the SAC hypothesis on the relationship between thickness of the spermatophore envelope and presence of signa in species differing in female mating patterns have not been tested. In this respect, the only relevant observations we are aware of are those reported by Matsumoto and Suzuki in a paper on mating plugs in six genera of Papilionidae (Matsumoto & Suzuki, 1995). We have discussed these data in detail in previous publications (Cordero, 2005; Sánchez, Hernández-Baños & Cordero, 2011). Briefly, Matsumoto and Suzuki’s results support predictions of the SAC hypothesis: monandrous genera are characterized by an absence of thick spermatophore envelopes (“capsule” in their terminology) and a lack of signum; moderately polyandrous species have a “relatively thick” spermatophore envelope and a “small” signum; whereas more polyandrous genera have a “thick” spermatophore envelope and a well developed signum (Matsumoto & Suzuki, 1995). The agreement of Matsumoto and Suzuki’s data with the SAC hypothesis is persuasive, but studies specifically designed to test the predicted relationship between the thickness of spermatophore envelopes and signa are necessary. In this report, we use data on the thickness of spermatophore envelopes of eleven species of butterflies varying in presence of signa and in female mating pattern (Fig. 2A) to test two predictions of the SAC hypothesis. First, we tested the prediction that spermatophore envelopes of polyandrous species with signa are thicker than those of monandrous species without signa (T1 → T2 in Fig. 1). Second, we tested the prediction that spermatophore envelopes of monandrous species with signa have thicker spermatophore envelopes than their closer polyandrous relatives with signa; as explained above, the rationale behind this prediction is that in monandrous species with signa monandry is enforced by males via the (co)evolution of “very thick” spermatophore envelopes (T2 → T4 in Fig. 1).

Figure 2 Comparative tests of the sexually antagonistic coevolution hypothesis (SAC) of the evolution of spermatophore envelope thickness in butterflies.

(A) Phylogenetic relationships between the eleven butterfly species included in the comparisons (this figure is part of the phylogenetic supertree used in the comparative study of Sánchez, Hernández-Baños & Cordero (2011). (B) Comparison of spermatophore envelope thickness between one polyandrous species with signa and two monandrous species without signa. As predicted by the SAC, the polyandrous species with signa has thicker envelopes than the monandrous species without signa. (C-E) Three comparisons of spermatophore envelope thickness between polyandrous species with signa and monandrous species with signa. As predicted by the SAC, in comparisons A and B the monandrous species with signa has thicker envelopes than polyandrous species with signa; this pattern was not observed in case C.

Materials and Methods

We collected females from eleven species of the subfamily Heliconiinae (Nymphalidae) (Luis-Martínez, Llorente-Bousquets & Vargas-Fernández, 2003; Table 1); specimens were captured under a scientific collector permit granted to the second author by the Mexican Secretaría de Medio Ambiente y Recursos Naturales (FAUT-0237). These species were selected to test the predictions mentioned in the introduction on the basis of findings from our previous research (Sánchez, Hernández-Baños & Cordero, 2011). Information about the absence/presence of signa was obtained from Brown (1981) and confirmed upon dissection. For most species, we used published data about female mating pattern estimated from spermatophore counts in field collected females (Heliconius spp. (Ehrlich & Ehrlich, 1978; Walters et al., 2012); Eueides spp., Dryadula phaetusa, Dryas iulia, Philaethria diatonica and Dione juno: (Ehrlich & Ehrlich, 1978); Agraulis vanillae (Drummond, 1984); Dione moneta: data from females collected by VS in the Pedregal de San Angel ecological preserve, located in the main campus of the Universidad Nacional Autónoma de México in southern Mexico City, these females were different from those used for measuring thickness of spermatophore envelopes). Most females were collected in different locations in the state of Veracruz, Mexico. Females were netted, euthanized, and their abdomens preserved in vials with 70% ethanol until dissection.

Table 1 Descriptive statistics of spermatophore envelope thickness (mm) of each spermatophore measured.

Each row corresponds to one spermatophore of the species indicated in the first column (total sample: 11 species and 43 spermatophores). ns: total number of measurements made in sections of each individual spermatophore (in almost all cases there were four measurements per section). Species with an asterisk are polyandrous, all the others are monandrous.

Species/Specimen	n s	Mean	SD	Median	Min.–Max.	
1. Heliconius ismenius*	227	0.034	0.010	0.03	0.01–0.06	
2. H. ismenius*	157	0.035	0.012	0.03	0.01–1.00	
3. H. ismenius*	315	0.033	0.009	0.03	0.01–0.07	
4. H. ismenius*	106	0.034	0.010	0.03	0.01–0.06	
5. H. ismenius*	154	0.037	0.010	0.04	0.02–0.06	
1. Heliconius hortense	188	0.028	0.009	0.03	0.01–0.05	
2. H. hortense	127	0.027	0.009	0.03	0.01–0.06	
3. H. hortense	179	0.028	0.008	0.03	0.01–0.05	
4. H. hortense	78	0.028	0.009	0.03	0.01–0.04	
5. H. hortense	163	0.031	0.007	0.03	0.02–0.05	
1. Heliconius charithonia	187	0.021	0.007	0.02	0.01–0.04	
2. H. charithonia	55	0.024	0.005	0.02	0.02–0.04	
1. Eueides aliphera*	123	0.039	0.009	0.04	0.02–0.06	
2. E. aliphera*	89	0.032	0.009	0.03	0.01–0.06	
3. E. aliphera*	102	0.036	0.008	0.04	0.02–0.05	
4. E. aliphera*	83	0.036	0.008	0.04	0.01–0.05	
1. Eueides isabella	136	0.047	0.015	0.05	0.02–0.09	
2. E. isabella	147	0.049	0.014	0.05	0.02–0.09	
3. E. isabella	232	0.060	0.018	0.06	0.02–0.12	
4. E. isabella	147	0.054	0.012	0.05	0.03–0.10	
5. E. isabella	93	0.052	0.015	0.05	0.03–0.09	
6. E. isabella	209	0.052	0.015	0.05	0.03–0.12	
7. E. isabella	248	0.069	0.027	0.06	0.03–0.16	
1. Dryadula phaetusa*	285	0.048	0.013	0.05	0.02–0.08	
2. D. phaetusa*	221	0.045	0.011	0.05	0.02–0.10	
3. D. phaetusa*	238	0.042	0.012	0.04	0.01–0.08	
4. D. phaetusa*	413	0.054	0.013	0.05	0.03–0.09	
5. D. phaetusa*	280	0.045	0.016	0.04	0.02–0.11	
1. Dryas iulia*	236	0.047	0.014	0.05	0.01–0.09	
2. D. iulia*	195	0.033	0.012	0.03	0.01–0.08	
3. D. iulia*	120	0.043	0.020	0.04	0.01–0.09	
1. Philaethria diatonica	272	0.069	0.018	0.07	0.03–0.12	
2. P. diatonica	333	0.070	0.018	0.07	0.03–0.12	
3. P. diatonica	316	0.063	0.022	0.06	0.02–0.21	
1. Agraulis vanillae*	248	0.047	0.011	0.05	0.02–0.08	
2. A. vanillae*	154	0.054	0.011	0.05	0.03–0.09	
3. A. vanillae*	184	0.053	0.018	0.05	0.02–0.14	
1. Dione moneta	226	0.032	0.008	0.03	0.01–0.06	
2. D. moneta	140	0.034	0.011	0.03	0.01–0.08	
3. D. moneta	219	0.037	0.013	0.04	0.01–0.10	
1. Dione juno	134	0.053	0.013	0.05	0.03–0.09	
2. D. juno	210	0.048	0.018	0.04	0.02–0.10	
3. D. juno	151	0.049	0.011	0.05	0.03–0.10	

In the laboratory, the corpus bursae were dissected under a dissection microscope (Olympus SZH10) and only corpus bursae containing complete spermatophores were cut in transversal sections that allowed us measuring the thickness of spermatophore envelopes. (Several females provided no data because they did not contain spermatophores or because the spermatophores they contained were partially or completely digested.) To obtain cross sections of spermatophore envelopes, the corpus bursae containing intact spermatophores were processed in the following sequence: (1) they were left in Bouin fixative solution for 24 h; (2) they were dehydrated in progressively higher concentrations of alcohol (from 50% to 100%, leaving the corpus bursae 1 h in each concentration); (3) they were left in a 1:1 mixture of Paraplast® tissue embedding media and HistoChoice® clearing agent for 24 h in an oven at 60°C; (4) they were left in Paraplast® tissue embedding media for 24 h in an oven at 60°C; (5) blocks of Paraplast® containing one corpus bursa were elaborated; (6) the whole corpus bursae containing intact spermatophores were cut transversally in 20 μm thick sections with an advanced precision rotary microtome (MD00030); (7) the sections were placed in glass slides, stained with methylene blue, and permanent preparations made using Cytoseal Mounting Medium. Photographs of these preparations were taken under the microscope (Olympus BX-51) with a digital camera (Olympus C-5050), and the thickness of spermatophore envelopes measured in the photographs of the sections with the UTHSCSA ImageTool for Windows version 3.00 software. In each photograph we traced an imaginary cross centered in the middle point of the section and measured the thickness of the spermatophore envelope at each of the four intersection points between the cross and the spermatophore section. The number of spermatophores used per species varied between 2 and 7 (total number of spermatophores studied = 43); the total number of measurements of envelope thickness per spermatophore varied between 55 and 413 (about half of the sample had between 150 and 250 sections measured), mainly due to differences in spermatophore size (Table 1).

The prediction that spermatophore envelopes of polyandrous species with signa are thicker than those of monandrous species without signa was tested by comparing three species of Heliconius, two belonging to the monandrous clade without signa (H. hortense and H. charithonia) and the other to the polyandrous clade with signa (H. ismenius) (Beltrán et al., 2007; Fig. 2A). The prediction that spermatophore envelopes of monandrous species with signa are thicker than those of their polyandrous relatives with signa was tested in three independent comparisons (Fig. 2A): (a) polyandrous Eueides aliphera vs. monandrous E. isabella; (b) polyandrous Dryadula phaetusa+Dryas iulia vs. monandrous Philaethria diatonica; and (c) polyandrous Agraulis vanilla vs. monandrous Dione juno+D. moneta.

Results and Discussion

Are the spermatophore envelopes of polyandrous species with signa thicker than those of monandrous species without signa?

The spermatophore envelopes of the polyandrous species with signa (H. ismenius) were thicker than those of the monandrous species lacking signa (H. hortense and H. charithonia) (Kruskal-Wallis ANOVA, H2,12 = 8.33, p = 0.016; Fig. 2B). This result is in agreement with the SAC hypothesis that proposes that polyandry selects for males that produce thicker spermatophore envelopes to delay female remating, and that, in response, females evolved signa that allowed them to increase the rate of spermatophore digestion, thus increasing their remating rate (Cordero, 2005; Sánchez, Hernández-Baños & Cordero, 2011). There were also differences in spermatophore envelope thickness between the two monandrous species (Fig. 2B). Since Walters and colleagues found that in large samples of some pupal mating monandrous Heliconius species there is a very small proportion of double mated females (Walters et al., 2012), it would be interesting to study large samples of H. hortense and H. charithonia to see if some females mate more than once and, in case they do, if the proportion of multiple mated females is larger in H. hortense, as would predict the SAC hypothesis.

Are spermatophore envelopes of monandrous species with signa thicker than those of their polyandrous relatives with signa?

In two of the three groups compared, the envelopes of the spermatophores received by monandrous females with signa were thicker than those of polyandrous species with signa (Eueides species [Fig. 2C]: Mann-Whitney Test, U = 0, p = 0.006; Dryadula/ Dryas/ Philaethria [Fig. 2D]: Kruskal-Wallis ANOVA, H2,11 = 6.91, p = 0.032). These results agree with expectations from the SAC hypothesis, that predicts perpetual coevolution between male and female traits and, therefore, considers the possibility of finding instances in which the interests of one of the sexes (males in the present case) prevail over those of the opposite sex (females in the present case), as would be the situation depicted in time 4 of Fig. 1. However, although these results are consistent with the prediction, they do not prove that in E. isabella and P. diatonica monandry is imposed by males and, therefore, maladaptive for females. To test this, it is necessary to show that females of these two species do not remate due to the time taken to break and digest the spermatophore, and that female fitness decreased when they lost the ability to remate due to the evolution of thicker spermatophore envelopes.

On the other hand, the third comparison (Fig. 2E) does not support the prediction: the thinner spermatophore envelopes were present in one of the monandrous species (Dione moneta), while the other (D. juno) had spermatophore envelopes as thick as those of the polyandrous species (Agraulis vanillae) (Kruskal-Wallis ANOVA, H2,9 = 6.23, p = 0.044). A hypothesis to explain this case is that selection favored monandry in female D. moneta, which, in turn, favored the evolution of thin spermatophore envelopes. However, if the reduction in envelope thickness evolves gradually, the decrease in signa size and/or in the size of the spines covering the signa (see next paragraph and Fig. 3) also could be gradual, and the presence of signa and a relatively thin spermatophore envelope could be expected as a transitory evolutionary state. It is interesting to note that, although thinner when compared with D. juno and A. vanillae, the spermatophore envelopes of D. moneta are thicker than those of the two monandrous Heliconius species without signa (Fig. 2B).

Figure 3 The length of the spines covering the signa correlates with spermatophore envelope thickness.

(A) Comparisons between the thickness of spermatophore envelopes and the length of the spines that cover the signa in two species selected for producing thick spermatophore envelopes, the polyandrous Heliconius ismenius and the monandrous Eueides isabella. (B) Section of signum covered with spines next to a section of the spermatophore envelope from a female H. ismenius. (C) Section of signum covered with spines next to a section of the spermatophore envelope from a female E. isabella. Photographs (B) and (C) taken from Galicia, Sánchez & Cordero (2008) with permission from The Entomological Society of America.

A final observation is consistent with the hypothesis of antagonistic coevolution between signa traits and spermatophore envelopes: In the polyandrous H. ismenius and the monandrous Eueides isabella, females have two signa shaped like long and thin plates covered with small spines (this general structure is present, with variants, in most species included in this paper), and previous observations indicate that these small spines help breaking open the spermatophore envelope (Galicia, Sánchez & Cordero, 2008). When we compared the thickness of the spermatophore envelopes with the average length of the spines covering the signa we found a good match between these two measures (Fig. 3). As the SAC hypothesis would predict, the spines are longer in the species with thicker spermatophore envelopes (E. isabella) and in both species they are of a length similar to the thickness of the spermatophore envelopes produced by males of its own species.

Conclusions

In general terms, most of the comparisons presented in this paper are consistent with the idea that sexually antagonistic selective pressures have been important forces in the evolution of female mating patterns, signa and spermatophore envelope thickness in heliconiinae butterflies (Cordero, 2005; Sánchez, Hernández-Baños & Cordero, 2011; Fig. 1): (a) The spermatophore envelopes of a polyandrous species with signa are thicker than those of two monandrous species without signa (Fig. 2B); (b) in two out of three cases, males from monandrous species with signa produced thicker spermatophore envelopes than related polyandrous species with signa (Fig. 2C, 2D); and (c) in two species the length of the spines covering the signa matched the thickness of the spermatophore envelopes produced by males of its own species (Fig. 3). On the other hand, one of the comparisons did not fit the prediction (Fig. 2E), and further studies are necessary to test if monandry is imposed by males in E. isabella and P. diatonica. When we consider that, at least in some species, signa could accomplish different or additional functions to spermatophore tearing (for example, protection from spines in male genitalia; (Galicia, Sánchez & Cordero, 2008; Cordero, 2010), it is not surprising that not all variation in the presence of signa and in spermatophore envelope thickness could be explained by the SAC hypothesis. Future comparative and functional studies are necessary to fully understand the evolution of these traits.

This work is part of VS doctoral thesis (Posgrado en Ciencias Biológicas, Universidad Nacional Autónoma de México). We thank Dr. Raúl Cueva del Castillo and Dr. Rogelio Macías for comments and suggestions to our research; two anonymous reviewers and Dr. Dezene Huber provided comments that helped us to improve the manuscript. We thank Raúl Martínez for technical help.

Additional Information and Declarations

Competing Interests

Author Contributions

The authors have no competing interests.

Víctor Sánchez conceived and designed the experiments, performed the experiments, analyzed the data, wrote the paper.

Carlos Cordero conceived and designed the experiments, analyzed the data, contributed reagents/materials/analysis tools, wrote the paper.

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
