# Peer review of "Sexual coevolution of spermatophore envelopes and female genital traits in butterflies: Evidence of male coercion?"

_PeerJ, doi:10.7717/peerj.247_

## Round 0.1 · original submission · Minor Revisions

Thank you for submitting this interesting study to PeerJ.

One reviewer recommended that the paper be accepted, although they also asked for what amount to be some very minor revisions.

The other reviewer recommended major revisions, but they are all that can be taken care of with some further literature review and adjustments to the text.

In particular, as suggested by one reviewer, I think that it would be worthwhile to explore the degree of polyandry in the various species as this could add further value to this paper. The other reviewer also touches on this when they say: "Authors should explain how the species were categorized as monandrous or polyandrous."

As neither reviewer is asking for more field or lab work, and since the reviews are both generally positive (or at least are not negative) in terms of the rigor of this study, I am recommending acceptance with minor revisions.

Please also note the slight typo in one reviewer's recommendations. Instead of "(f)igures 4 and 4 could be combined into one single figure..." the sentence should read "(f)igures 3 and 4 could be combined into one single figure..."

Please also keep in mind that PeerJ recommends open peer review. It would be great if you would release the peer reviews and your rebuttal to appear alongside the MS. This is useful for readers and can also be instructive for students who are learning the ropes of scientific publishing.

Reviewer 1 ·

Basic reporting

The paper adheres to all of Peer J policies

Experimental design

The experimental design is satisfactory. However, as I state in my review, I would welcome more information about the degree of polyandry among polyandrous species, rather than dichotomizing species into either "polyandrous" or "monandrous" - by doing this more detailed analysis of Sexualla Antagonistic Coevolution would be possible.

Validity of the findings

In this paper the authors test the hypothesis that spermatophore envelopes (produced by males) coevolves with the structure known as ”signa” in the female reproductive tract (a structure that ruptures the envelope of the spermatophore transferred by the male to the female during copulation). The idea is a straightforward example of potential Sexually Antagonistic Coevolution (SAC); males that have successfully mated with a female benefit from delaying female remating (by prolonging the time it takes for a female to rupture the envelope of the spermatophore), and females benefit from being able to rupture the envelope of the spermatophore as quickly as possible to be able to remate. Hence one prediction from the SAC hypothesis is that the spermatophore envelope should be thicker in polyandrous species with signa than in monandrous species without signa. This prediction is shown conclusively to hold by the empirical data provided in this paper.

The authors also derive a second prediction from the SAC hypothesis in this paper – that the spermatophore envelope in monandrous species WITH signa should be thicker than in polyandrous species with signa. This prediction is based on the hypothesis that, in these species, the ”battle of the sexes” has been won by the males, and that female monandry in these species is enforced by the males. I have problems with accepting this conclusion without other supporting evidence, and, moreover, this prediction is supported in only two of three comparisons, which cannot be regarded as strong support for the prediction itself. Nonetheless the paper is written as if the data support the prediction without any substantial discussion. This is unsatisfactory.

I also would welcome more information on the degree of monandry/polyandry in the heliconiid butterflies analysed in this paper. Dichotomising species as monandrous or polyandrous is not very satisfactory, and, moreover, by providing more detailed data on the degree of polyandry the authors would be able to test yet an additional prediction from the SAC hypothesis – that the envelope of the spermatophore would increase with the degree of polyandry.

Although I like the approach and work that is presented in this paper I think that the authors need to discuss their results in a more ”relaxed” manner, as it now stands I they interpret all their results as supporting their predictions, an interpretation that invites more scepticism in the reader’s mind than convinction.

Additional comments

As I state above, I find the paper interesting, but it must be written in a way that consider both in what way the data support their predictions and in what way they do not. The way the paper is now written the authors enthusiastically write as if all of their findings support their predictions, and I think that they need to "problematize" their findings more

Reviewer 2 ·

Basic reporting

Figure 1 is very simple and it is not needed given the clear explanation provided in the text.
Figure 2 shows data published elsewhere and may not be needed.
Figures 4 and 4 could be combined into one single figure, with points paired as in the statistical analyses performed. This would help the readers compare the entire range of variation across the sampled Heliconiinae species.

Experimental design

Authors should explain how the species were categorized as monandrous or polyandrous.
Heliconius erato was not sampled. Why?

Validity of the findings

Although chemical analyses have never been performed, males of all Heliconiinae examined have the potential to deliver anti-aphrodisiac to the females (and all females have stink clubs). This should be discussed in more detail given that it is relevant to the results and conclusions.

Additional comments

The data is interesting and well presented. The text is very clearly written.

---

## Round 0.2 · accepted · Accept

Thank you for your clear and precise response to the reviews. Thank you, also, for being willing to publish the reviews and your rebuttal alongside the paper. Those will provide a great deal of context for readers.

And, of course, thanks to the referees for their excellent reviews of this MS.